# Fairness-Based Multi-AP Coordination Using Federated Learning in Wi-Fi 7

**DOI:** 10.3390/s22249776

**Published:** 2022-12-13

**Authors:** Gimoon Woo, Hyungbin Kim, Seunghyun Park, Cheolwoo You, Hyunhee Park

**Affiliations:** 1Department of Information and Communication Engineering, Myongji University, Yongin 17058, Republic of Korea; 2Division of Computer Engineering, Hansung University, Seoul 02876, Republic of Korea

**Keywords:** federated learning, distributed machine learning, multi-AP coordination, energy consumption optimization, 802.11be, Wi-Fi 7

## Abstract

Federated learning is a type of distributed machine learning in which models learn by using large-scale decentralized data between servers and devices. In a short-range wireless communication environment, it can be difficult to apply federated learning because the number of devices in one access point (AP) is small, which can be small enough to perform federated learning. Therefore, it means that the minimum number of devices required to perform federated learning cannot be matched by the devices included in one AP environment. To do this, we propose to obtain a uniform global model regardless of data distribution by considering the multi-AP coordination characteristics of IEEE 802.11be in a decentralized federated learning environment. The proposed method can solve the imbalance in data transmission due to the non-independent and identically distributed (non-IID) environment in a decentralized federated learning environment. In addition, we can also ensure the fairness of multi-APs and determine the update criteria for newly elected primary-APs by considering the learning training time of multi-APs and energy consumption of grouped devices performing federated learning. Thus, our proposed method can determine the primary-AP according to the number of devices participating in the federated learning in each AP during the initial federated learning to consider the communication efficiency. After the initial federated learning, fairness can be guaranteed by determining the primary-AP through the training time of each AP. As a result of performing decentralized federated learning using the MNIST and FMNIST dataset, the proposed method showed up to a 97.6% prediction accuracy. In other words, it can be seen that, even in a non-IID multi-AP environment, the update of the global model for federated learning is performed fairly.

## 1. Introduction

Wireless local area networks (WLANs) that conform to the IEEE 802.11 standard are the most widely used networks in the world and are commonly referred to as wireless fidelity (Wi-Fi) networks. In May 2019, the IEEE 802.11 working group established a task group to present a range of next-generation WLANs, including IEEE 802.11be, referred to as Wi-Fi 7 [1]. The maximum data rate for Wi-Fi 7 is expected to be 46 Gbps, and developers aim to achieve high-throughput by far exceeding the 9.6 Gbps maximum data rate of IEEE 802.11ax (Wi-Fi 6). To achieve this goal, the IEEE 802.11 working group is currently discussing proposals that include multi-link operation (MLO), channel sounding, and multi-access point (multi-AP) coordination.

Wi-Fi 7 has been proposed for various new applications, such as those using 4 K/8 K video resolution, which require a high transmission speed and low latency. Federated learning (FL) is emerging as a potential way to address the increasing number of connected devices possessed by individuals. To implement FL, it is necessary to select a device in an idle state that can participate with its small storage space or that requires low computational power [2]. In WLAN, there is lower coverage than in cellular methods such as 4G and 5G, and fewer devices are involved. This makes it difficult to select devices that can participate in FL to derive a generalized global model.

The system model-based FL does not involve the process of collecting raw data from servers, unlike general deep learning methods such as a convolution neural network (CNN) model [3]. Instead, FL learns while sharing a model or weight between a server and a device, in which the data of individual devices are preserved. Existing FL systems assume a cloud-based system structure. However, this structure is causing communication latency problems between the clients and servers. In addition, because the performance of the overall FL system depends on the server performance, so the larger the environment, the lower its efficiency. Hierarchical FL has been proposed to address this problem [4]. Hierarchical FL involves the aggregation of the global model on a connected edge server rather than on a central server. This can reduce the cost of server-to-device communication and overcome problems related to cloud-based structures that rely on server performance in large environments. In addition, a decentralized FL approach has also been proposed by another study [5]. Because decentralized FL does not require a central server, each device may conduct FL without communicating with a server. In this manner, it is not necessary to consider the cost of server-to-device communication.

In this study, we propose a method for a multi-link operation using an FL algorithm in a Wi-Fi 7 environment. Since the WLAN environment has less coverage than the cellular environment, FL is performed with fewer devices. To this end, the proposed method creates an FL model suitable for the WLAN environment by fairly sharing the global model of FL using multi-APs. By considering the characteristics of a Wi-Fi 7 multi-AP, each neighboring AP can train the local models of all neighboring devices by transmitting the weights of the local model of the connected devices [6]. A benefit of this approach is that all clusters can have uniform data regardless of the data distribution, i.e., a generalized global model. Moreover, as opposed to conventional cloud-based FL, sharing models between APs can result in reduced communication costs. To this end, multiple APs that may be included in a common large cluster are considered multi-APs. In addition, the multi-APs contained in the common cluster are classified into primary-AP (P-AP) and secondary-AP (S-AP) in the proposed method, respectively. In addition, for fairness in this multi-AP operation, the P-AP is updated periodically to change the role with the S-AP. To periodically update the P-AP in the proposed Wi-Fi 7 multi-AP coordination scheme, an AP-specific energy consumption is considered. Therefore, we propose a new method for creating and exchanging a global optimized model in the Wi-Fi network in which the terminal operates by proposing the FL model through multi-AP coordination in Wi-Fi 7. For a Wi-Fi network that supports the dynamic network operation of a terminal, the P-AP and S-AP are determined according to the number of devices participating in FL within each AP, and we propose an algorithm that exchanges roles with the P-AP and S-AP according to the training time. The main contributions of this study are as follows:(1)A method for periodically determining P-AP updates during FL is proposed, which considers the signal-to-noise ratio (SNR) and energy consumption of each device.(2)A high-performance FL system without a central server is proposed, in which each AP with non-uniform data distribution in decentralized FL shows an increase in the prediction accuracy when predicting new data.

The rest of this study is organized as follows: In Section 2, we introduce studies that overcome the limitations of cloud-based structures in FL, and a study on Wi-Fi 7 introduction and multi-AP coordination. Section 3 describes the proposed method. Section 4 describes and discusses the simulation results using the proposed method. Conclusions are provided in Section 5.

## 2. Related Work

FL is a machine learning method that learns distributed data from personal devices such as smartphones [7]. To proceed with FL, the server must first determine the devices that will participate in the FL process. The determinants are that the device is idle or charged and connected to a network such as Wi-Fi, i.e., if the device capability is less than the server in terms of battery life and data storage, it cannot participate if it is negatively affected learning.

As opposed to conventional FL, in which local models are aggregated and averaged on cloud servers, hierarchical FL aggregates local models on each edge server or AP that is connected to the device. Based on hierarchical FL, a method has been proposed in which devices within the coverage provided by overlapping neighboring edge servers share data from each edge server, thereby learning a global model without reaching a cloud server [8,9]. If a large number of devices is in overlapping areas, they perform well. However, this approach presents the limitation of device battery and performance degradation in overlapping areas in a 5G environment. In addition, communication efficiency between the device and edge server should be considered in a large-scale environment.

To overcome these limitations, several studies have been conducted to overcome communication efficiency through decentralized FL [10,11]. Decentralized FL is a peer-to-peer approach that does not depend on a central server, and each group learns and shares the local model to update the global model. An independent group can be a cloud server or an AP. In [11], the segmented gossip approach is proposed to update the global model by sharing weights during federated learning and fine-tuning parameters for each round in a decentralized FL system. As shown in the simulation results, the segmented gossip approach shows a higher accuracy than the existing federated learning model, but the process for each group to perform a global update after a local update is omitted. However, this process can differ in accuracy depending on the amount of training of FL or the number of devices and groups. To update the global model, the global model update is considered as the training time based on the energy consumption through the P-AP.

With the progress of next-generation WLAN, IEEE 802.11be (named Wi-Fi 7) is the process of standardization. As opposed to conventional WLAN schemes, such as IEEE 802.11ax (named Wi-Fi 6 and Wi-Fi 6e), multi-link operation and multi-AP coordination are proposed as new schemes for use in Wi-Fi 7. These allow for the collaboration between neighboring APs for required scheduling information. Various architectures that adopt these schemes have been proposed [12,13,14]. For example, one architecture shares information between adjacent APs by dividing the P-AP and S-AP roles of each AP, as shown in Figure 1. In the first phase, a multi-AP coordination set from neighboring APs (which could be S-APs) is configured by broadcasting a beacon signal in the P-AP. In this case, the P-AP should also inform the neighboring AP whether it is possible to perform a function to participate in multi-AP coordination. In addition, neighboring APs (which could be S-APs) that receive signals must also have a multi-AP coordination function. In the second phase, an adjacent AP (which could be an S-AP) receiving a signal may configure a multi-AP coordination set by transmitting a response signal to the P-AP. Finally, the multi-AP coordination set may then share information with the device that is connected to the adjacent S-AP through the P-AP. In Figure 1, the first P-AP is handled by the first AP that attempts to configure a multi-AP coordination set with the neighboring APs. AP 1 sends a beacon signal from AP 2 and AP 3 to form a multi-AP coordination set, and AP 2 and AP 3 among the neighboring APs participate in the multi-AP coordination. Among the neighboring APs, AP2 and AP 3 have a multi-AP coordination function, and the devices (A, B, C, D, E, and F) connected to each AP can share data through the P-AP.

In [15], channel assignment (CA) is specified to overcome what causes topological changes that affect the routing decisions made at the routing layer, leading to unreliable networks. Accordingly, an inter-layer approach is required to ensure a reliable connection, so DJ-CAR (Distributed Jamming Resilient Channel Assignment and Routing) is proposed by optimizing the network path cost for a formulated cross-layer problem. DJ-CAR is a dynamic, distributed CA approach and can be applied in either flat or hierarchical architectures. DJ-CAR also has the advantage of being able to mitigate malicious jamming and avoid external interference through SNR and the number of neighboring devices.

In [16], to improve the network capacity and flexibility of MANET through multiple interfaces of STAs operating on multiple channels and overcome the jamming problem, a distributed and heuristic channel assignment algorithm called Channel Assignment and Jammer Mitigation (CA-JAM) was developed. The CA-JAM algorithm allocates a unique channel for every individual interface of one STA, and all STAs exchange assignment information through the beacon frames of all the interfaces. Therefore, each STA can use the table distributively to reduce the number of neighboring STAs using the same channel to avoid interference and, consequently, to improve throughput. CA-JAM is completely deployable without the use of control channels and central entities, improving connectivity while mitigating the jamming of MANETs. Since the proposed paper is a decentralized FL environment, it may be vulnerable to jamming attacks or security. In addition, when forming a multi-AP coordination group, a beacon signal is sent; therefore, the advantages of DJ-CAR and CA-JAM can be utilized. In addition, MANET is a system with free connectivity between nodes, and each node functions as a router and has the advantage of being free to implement a dynamic network topology. However, CA-JAM was specifically designed for MANET. FL is a large-scale distributed machine learning system that processes distributed data and requires high-throughput. Since MANET is an impromptu network, it has unstable link characteristics, and it is difficult to process distributed data or increase the value of the FL model. Therefore, large-scale federated learning can be conducted by organically sharing distributed data with adjacent APs through the multi-AP coordination method.

Because the FL system requires high-throughput, it is assumed that each device is idle for processing data immediately or connected to fixed Wi-Fi environments. To solve the high resource cost, Nguyen, et al. proposed to minimize the energy and channel cost while maximizing the number of global model transmissions in the Wi-Fi channel [17]. The deep Q-Learning (DQL) algorithm based on the deep-Q-Network (DQN) is proposed to determine the optimal energy and channel for FL without prior knowledge of the network. This is a higher reward than the existing algorithm, so it shows a better average utility performance, but this algorithm is needed to go through a large number of episodes for learning. After all, we aim to generalize the global model of federated learning through a small number of devices and a small amount of learning by using the multi-AP coordination of Wi-Fi 7.

## 3. Fairness-Based Multi-AP Coordination Method

In this section, the proposed Wi-Fi 7 multi-AP coordination method employs a decentralized FL system. The proposed method determines the generalized global model even in an environment that contains few devices, and we can use multi-AP coordination to share FL model weights among independent neighboring APs. The proposed method in this study is described by the architecture shown in Figure 2. In Figure 2, t represents the round of FL. In the initial one-time FL, the AP with the smallest number of devices is determined as the P-AP. Each AP has the different training time for the local model and energy consumption depending on the various factors such as the number of devices and performance of each AP. Therefore, the AP which has the longest training time according to the energy consumption is determined as the P-AP in the next round t+1.

We describe the proposed method through the following algorithm. The process described in Figure 2 is similar to a general FL system, from the perspective of FL. However, for the P-AP determination, each device is identified to participate in the initial round of FL. As combined learning progresses, the P-AP can be determined by considering the training time and energy consumption of each AP from time t+1 (the training second communication round 2 after initial training round t). The algorithm used in this study is shown in Algorithm 1.
**Algorithm 1:** Proposed fairness-based multi-AP coordination methodInput  Initialized model $w_{0}$Output  Final global model $w^{G}$1:*P-AP* $\mathrm{decision}\;P_{0}\leftarrow minD_{K_{M}}$;2:**for** each step t=1, 2, …, T **do**3:   **for** each device k=1, 2, …, K in parallel **do**4:$\;w_{k}\left(t\right)=w_{k}\left(t-1\right)-\eta \nabla F_{k}\left(w_{k}\left(t-1\right)\right)$;5:$\;w_{k}^{L}\left(t\right)\leftarrow \;\mathrm{Local}\;\mathrm{Update}\left(t,\;w_{k}\left(t\right)\right)$;6:$\;P_{t}^{m}\leftarrow S_{t}^{m}\;’s\;\;w_{k}^{L}\left(t\right)$;7:      
**end for**
8:     Calculate total energy consumption:9:$\;\mathrm{Energy}\;\mathrm{consumption}\;E_{cmp,\;\tau }^{t}$;10:$\;\mathrm{Communication}\;\mathrm{energy}\;E_{com,\;\tau }^{t}$;11:      Calculate the training and evaluation time for each AP:12:         Check the AP with the longest training time;13:$\;\mathrm{Select}\;P_{t+1}\;\mathrm{for}\;\mathrm{next}\;\mathrm{round}\;\mathrm{of}\;\mathrm{FL}$;14:$\;w^{G}\left(t\right)\;\leftarrow \;\mathrm{AGGREGATION}\left(k,w_{k}^{L}\left(t\right)\right)$;15:$\;\frac{1}{N}\sum_{k=1}^{K}N_{k}F_{k}\left(w\right)$;16:$\mathrm{Send}\;w^{G}\left(t\right)\;\mathrm{to}\;S_{t+1}^{m}$;17:**end for**

### 3.1. Initial Setup for FL

We describe an initial set of proposed methods. The initial setup is limited to Round 1 (Round 1), such as Algorithm 1, Line 1. First, devices participating in FL should be in a charge-idle state and should not be used. In this study, it is assumed that device mobility is not high when determining device candidates for FL participation. Accordingly, a candidate set *D* of the device capable of participating in FL may be configured. Configuring a candidate set capable of participating in FL prevents the battery loss of devices resulting from the random determination of the FL participating devices.
(1)$$d_{1},d_{2},\cdots ,d_{k_{m}}\in D_{K_{M}}\geq \;\gamma ,$$
where γ is assumed to be an IEEE 802.11ax MCS 11-based threshold. MCS 11 is simply assumed to have the maximum channel environment and is named the lowest noise level. Devices with SNRs greater than γ may be included in set D, and the AP index is defined as 1,2,…,m∈M. In the initial setup step, the AP with the fewest elements among $D_{k_{m}}$ is set as the P-AP. In algorithm 1, the P-AP is defined as P and the S-AP is defined as S.

As shown in Algorithm 1, lines 2–9, the number of devices participating in the FL system within the range of each AP’s M is defined as $K_{M}$, and the total number of devices is set to $K=\sum_{m=1}^{M}K_{m}$. K is the total number of devices participating in FL. Because this study is based on a decentralized FL environment, the AP index is classified. Therefore, it is possible to distinguish which random device is connected in each AP through the AP index. When the number of data samples of the device k is defined as $N_{k}$, the total training data sample size is $N=\sum_{k=1}^{K}N_{k}$. The purpose of the FL system is to solve the following optimization problem [18].
(2)$$\underset{w}{\min}F\left(w\right)=\frac{1}{N}\overset{K}{\underset{k=1}{\sum }}N_{k}f_{k}\left(w\right).$$

Equation (2) minimizes empirical loss through the use of a training dataset in combined learning. N is the total training data sample, and $N_{k}={\left\{x_{i},\;y_{i}\right\}}_{i=1}^{N}$, where w is a vector that each device fully parameterizes while learning training data samples. $N_{k}$ is consistent with $x_{i}$ and $y_{i}$, where $x_{i}$ is the i th input sample, and $y_{i}$ is the i th response label. Here, $f_{i}\left(w\right)=f\left(w,x_{i},\;y_{i}\right)$ is a local loss function for the i th data sample. To solve the loss function, FL algorithms such as FedAvg [19] allow the devices participating in FL to obtain a global model without sharing personal data.
(3)$$w_{k}\left(t\right)=w_{k}\left(t-1\right)-\eta \nabla F_{k}\left(w_{k}\left(t-1\right)\right),$$
where $F\left(w\right)$ depends on the communication environment or machine learning model, which may be solved using the gradient descent. In Equation (3), the time step $t\left(t=1,\;2,\;\ldots ,\;T\right)$ is the update step expressed as a gradient descent size. For the local updating of each AP, the model parameters of the device k may be updated. The P-AP then receives a local updated model $w^{L}$ from the S-APs and aggregates all models to generate a global model $w^{G}$. The global loss function is described by:(4)$$F\left(w\right)=\frac{1}{N}\sum_{k=1}^{K}N_{k}F_{k}\left(w\right).$$

### 3.2. Primary AP Decision with Energy Consumption

After the initial setup phase in decentralized FL, criteria are required to aggregate the global model. During the FL process, the FL training times of each AP may differ for various reasons. For example, the training time varies according to the amount of data of each AP, or the number of connected devices. In addition, because a device can assume minimal mobility, the forms of data distributed in each AP differ. This study, therefore, considers the energy consumption of grouped devices for FL participation connected to each AP according to the FL training time for efficient FL [20]. This is performed to consider the fairness between APs and to increase the communication efficiency during the FL process, because APs cannot provide the high performance of a cloud server. The total energy consumption may be divided into local computing energy, energy consumption $E_{cmp,\;\tau }^{t}$, and communication energy $E_{com,\;\tau }^{t}$ as follows:(5)$$E_{cmp,\;\tau }^{t}=\frac{\alpha_{\tau }^{t}C_{\tau }^{t}{\left(F_{\tau }^{t}\right)}^{2}}{2},\;$$
(6)$$E_{com,\;\tau }^{t}=T_{\tau }^{t}P_{\tau }^{t}.$$

In Equation (5), τ is a communication round, $\alpha_{\tau }^{t}/2$ is the calculated capacity coefficient for each device equipped with different chipsets, $C_{\tau }^{t}$ is the period required for each device to execute a repetition once in the AP, and $F_{\tau }^{t}$ is the CPU period frequency of each device. In Equation (6), $T_{\tau }^{t}$ is the allocated communication time fraction and $P_{\tau }^{t}$ is the power function of each device. Downlink energy is not considered because the power of the parameter server is higher than the transmission power of the device, and the downlink time can be ignored compared to the uplink time. As shown in Figure 2, the AP with the longest training time for each communication round is selected as the next P-AP, and the P-AP is determined using the energy consumption during the FL process, except for that of the first round. With the exception of the APs with the longest training times, the other APs prepare for local model transmission by sending signals periodically to the adjacent APs after local training in the current round. Information on APs that have not been learned in the first round is available to all APs that participate in the multi-AP coordination set. When the learning is finished, the P-AP sends a local model to the AP that has the longest learning time. The corresponding AP is determined as a P-AP in the next round, and, after aggregation, a global model is transmitted to neighboring APs.

## 4. Simulation Result and Discussion

### 4.1. Simulation Parameters

This section describes the simulation parameters and results obtained in this study using the proposed method, and our experimental data, using the MNIST [21] and FMNIST [22] datasets. The MNIST and FMNIST datasets are split into 60,000 training samples and 10,000 test samples. In addition, 29,854 evaluation samples are used. These datasets are described in Table 1.

The LeNet-5 model [21], utilized deep neural network, set 1 local iteration, and 100 communication rounds are used in this study. In addition, set 50 and 100 FL participating devices are used, and each device has one random “target” label to highlight the non-uniformity of the data. For example, device 1 may have a labeled-”3” image of the MNIST dataset, and device 2 may have a labeled-”2” image of the MNIST dataset. When performing FL with only a small number of devices, the accuracy of the new prediction approach is derived using the least distributed labeled-”9” image of the MNIST dataset. Five and ten APs are set based on the lowest noise level, where each AP has one or more devices. A total of 50 or 100 devices are randomly divided and then connected. Each AP has a different number of devices, and the labels of each device in the APs differ; therefore, it may be seen that the data distribution is non-uniform. The example simulation diagram is described in Figure 3. Figure 3 is a schematic diagram of the simulation for this study. In [9], if AP 1 is determined to be a P-AP, a multi-AP coordination set can be configured with the neighboring AP 2 and AP 3. The APs in a multi-AP coordination set can share data through a wireless link. Device A to device F have random labels in the dataset, and each device is randomly connected to each AP.

Each device has a random label in the dataset, and each device is randomly linked to each AP. To configure the multi-AP coordination sets, an AP to become a P-AP sends a beacon signal and is connected through a wireless link.

### 4.2. Simulation Results

The predictive accuracy of the proposed method is compared with that of a method in which each AP learns individually, without sharing a global model in a decentralized FL system. Figure 4 shows a performance comparison using the MNIST dataset and FMNIST dataset, where M=5 (5 APs) and K=50 (50 devices) or K=100 (100 devices).

Figure 4 shows the accuracy obtained when FL is conducted for 50 and 100 devices, respectively, in an environment of five APs. In Figure 4, ‘50 devices using the MNIST dataset’ and ‘50 devices using the FMNIST dataset’ show the accuracy obtained using the MNIST and FMNIST datasets in the environments of 50 devices, and ‘100 devices using the MNIST dataset’ and ‘100 devices using the FMNIST dataset’ in Figure 4 show the accuracy of FL in the environments of 100 devices. When FL is using the MNIST dataset, the accuracies of the proposed method and comparative methods are 97.6% and 96.8% for 50 devices, and 96.1% and 95.9% for 100 devices, respectively. When using the FMNIST dataset, the accuracies of the proposed method are 86.0% for 50 devices and 84.3% for 100 devices. For comparison, the accuracy of the method without multi-AP coordination tends to increase gradually, without reaching convergence. The FMNIST dataset was created in 2017 to perform more demanding classification tasks than the MNIST dataset, which can easily achieve about 99% performance. Therefore, the convergence was not caused by the non-IID situation in this study’s experiment.

As in the case of general FL, the prediction accuracy values obtained using the proposed method converge. As shown in Figure 5, for the MNIST dataset, both the proposed method and comparative model show convergence. However, when the FMNIST dataset is used, the comparative model’s loss does not converge. Even if the accuracy of the model is as high as that shown in Figure 4, these results should be verified to check the predictive loss shown in Figure 5. Each independent AP has a random device and label, and, therefore, the comparative model is highly accurate, although it is more likely to make incorrect predictions. This implies that each AP has a non-uniform data distribution, as shown in Figure 5. These results also show that the loss graph for FL converges using the proposed multi-AP coordination method. It is confirmed that each independent AP generalizes to a non-IID situation through data sharing.

Figure 6 shows the accuracy of FL in environments with ten APs. As opposed to Figure 4, each of the ten APs contains randomly assigned devices with a random label. Therefore, as the number of APs increases, the amount of data in the APs decreases. Accuracy converges slowly when 50 devices are employed. However, the model accuracy follows a more stable trend than that shown in Figure 4 when 100 devices are used in an environment with five APs. Therefore, in the case of the MNIST dataset, the accuracy comparative and proposed method is 95.0% and 92.1% for 50 devices, and 97.7% and 97.6% for 100 devices, respectively. For the FMNIST dataset, the accuracy of the proposed method is 84.9% for 50 devices and 87.6% for 100 devices. As shown in Figure 6, we can confirm that the accuracy provided by using a ten-AP environment is 1.6% higher for the MNIST dataset and 3.3% higher for the FMNIST dataset than that of the environments containing five APs and 100 devices. Figure 7 shows a loss graph for FL using the MNIST and FMNIST datasets in M=10 environments with K=50 and K=100. As in the case of the environments with five APs, the MNIST dataset converges for both the proposed method and comparative method without the multi-AP coordination method. However, it is confirmed that the comparative method does not converge when the FMNIST dataset is employed. We believe that non-convergence can be solved by changing several parameters; for example, by increasing the number of devices or increasing the amount of learning.

Figure 8 shows the simulation result of confirming the convergence point of FL according to the channel environment. To show the possibility of FL with only the small number of devices, it is necessary to confirm that the FL model is generalized according to the channel environment. In other words, the accuracy should converge even if the channel environment is not good. In Figure 8, the ‘Multi-AP’ label is a good channel environment as the proposed method. The label ‘Noise_Multi-AP’ is a poor channel environment in comparison. Noise was intentionally added when each device transmits to the connected AP after a local update for the insignificant channel environment setting. The data error rate increases as the channel environment deteriorates due to the high noise level. In the case of the MNIST dataset, the result is converged after about 100 communication rounds in the good channel environment and is converged after about 150 communication rounds in the poor channel environment. In the case of the FMNIST dataset, the result is converged after about 150 communication rounds in the good channel environment and is gradually converged from about 150 communication rounds in the poor channel environment.

Figure 9 shows the comparison in operation time between the system that determines the P-AP as the AP with the longest training time proposed in this study and the system that determines the P-AP as the AP with the shortest training time. In Figure 8, the simulation is conducted in ten AP environments. In the environment of 50 devices, as the running time increased, it shows a difference of about 1.39 s for the MNIST dataset and 0.03 s for the FMNIST dataset. On the other hand, in the environment of 100 devices, the difference is about 1.65 s for the MNIST dataset and about 1.56 s for the FMNIST dataset. The operation time for the number of devices can reduce the time cost incurred in determining the P-AP at every running time to ensure fairness. In addition, it is confirmed that the effect is proportional as the number of devices increases in Figure 9.

## 5. Conclusions

In this study, fairness-based multi-AP coordination using FL architecture is proposed to derive a generalized global model with a small number of devices by applying the multi-AP coordination of Wi-Fi 7. The proposed method exhibits FL effects in cellular environments in short-range wireless environments with low coverage. The proposed method can reduce the overall communication costs of a system by selecting an FL-participable device and considering energy conservation during the FL process. In addition, a new method to determine the primary-AP in a multi-AP coordination is proposed, which considers the energy consumption for FL. By considering the energy consumption, fairness between APs is ensured during the FL process. In other words, the AP generally demonstrates a lower performance than the cloud server. Therefore, a consideration of fairness for the P-AP decision by using the FL training time and energy consumption can reduce the communication cost of FL applications that require a large capacity. The simulation results obtained in this study indicate a higher accuracy and convergent losses of the proposed method compared to cases in which multi-AP coordination was not applied. The results of this study show that this FL-based approach could be used in the further development of next-generation Wi-Fi 7 applications. In this study, a method is proposed to determine the AP with the fewest devices participating in FL as the P-AP in the initial setting of FL, and an efficient FL algorithm is proposed by determining the AP with the longest training time as the next P-AP through the training time of each AP during FL. However, multi-AP transmission technologies, such as coordinated spatial reuse or joint transmission, are not applied in the multi-AP coordination of Wi-Fi 7. In the future, we will consider multi-AP transmission and further develop the proposed FL method to efficiently reduce communication costs, even in a significantly crowded environment.

## Figures and Tables

**Figure 1 sensors-22-09776-f001:**
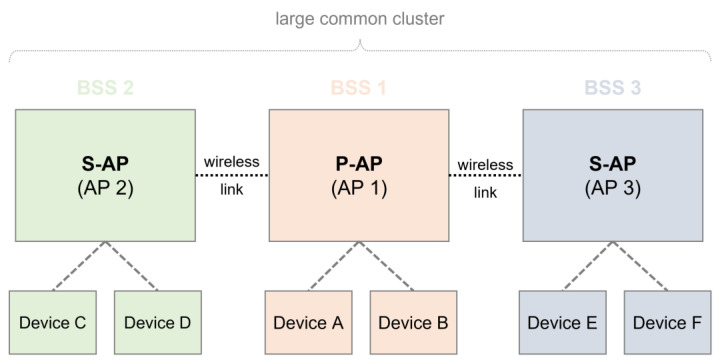
Multi-AP coordination in Wi-Fi 7.

**Figure 2 sensors-22-09776-f002:**
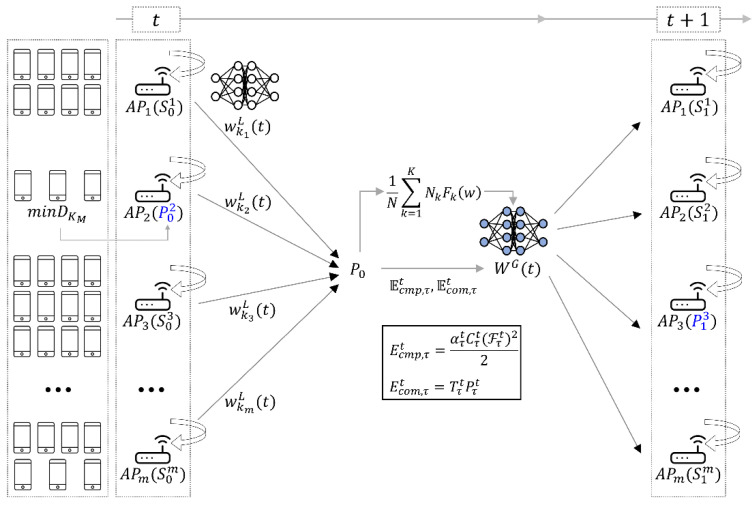
Architecture of the proposed method. The blue *P* is P-AP of the communication round. The superscript of *P* means the index of the AP, and the subscript is communication round.

**Figure 3 sensors-22-09776-f003:**
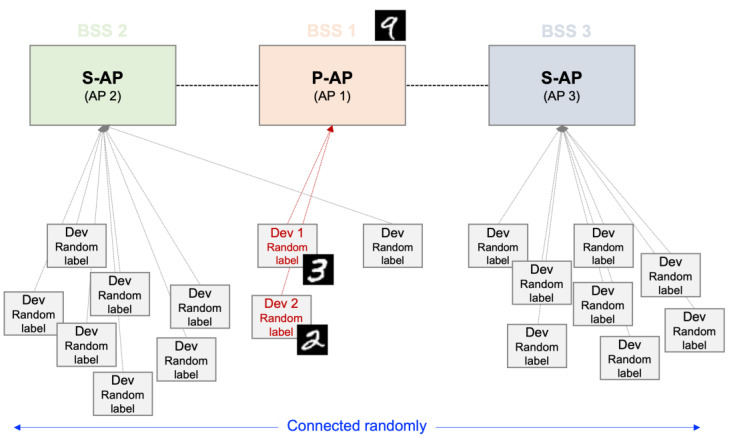
Example of simulation environment. For example, device 1, as Dev 1, has a labeled-“3” and devices 2, as Dev 2, has a labeled-“2” image of the MNIST dataset. To evaluate the accuracy of the new prediction is used the least distributed labeled-“9” image of the MNIST dataset.

**Figure 4 sensors-22-09776-f004:**
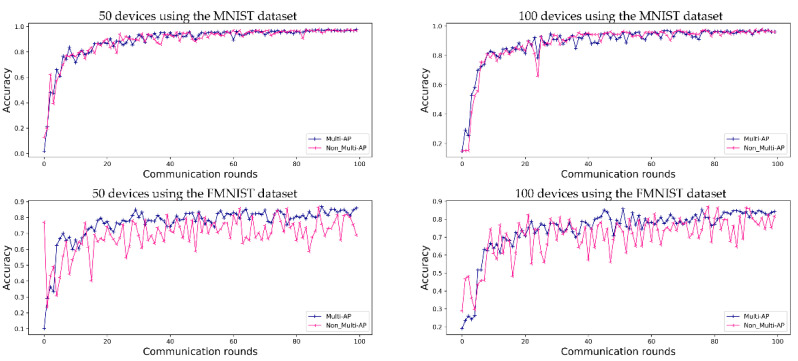
Evaluation of FL prediction accuracy using the MNIST and FMNIST datasets in an environment with five APs during 100 communication rounds. ‘Multi-AP’ shows the results for the proposed method, and ‘Non_Multi-AP’ shows those of a comparative method without multi-AP coordination.

**Figure 5 sensors-22-09776-f005:**
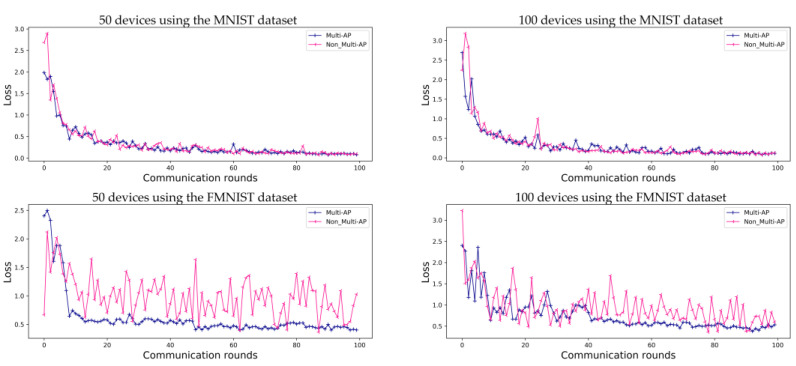
Evaluation of loss of prediction for FL using the MNIST and FMNIST datasets in environments with five APs, for 100 communication rounds. Both the proposed ‘Multi-AP’ and ‘Non_Multi-AP’ models converge when using the MNIST datasets, while the ‘Non_Multi-AP’ model does not converge when the FMNIST dataset is used.

**Figure 6 sensors-22-09776-f006:**
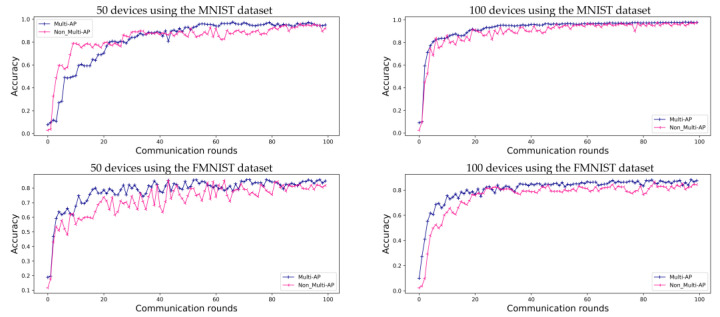
Evaluation of FL prediction using the MNIST and FMNIST datasets in environments with ten APs, for 100 communication rounds.

**Figure 7 sensors-22-09776-f007:**
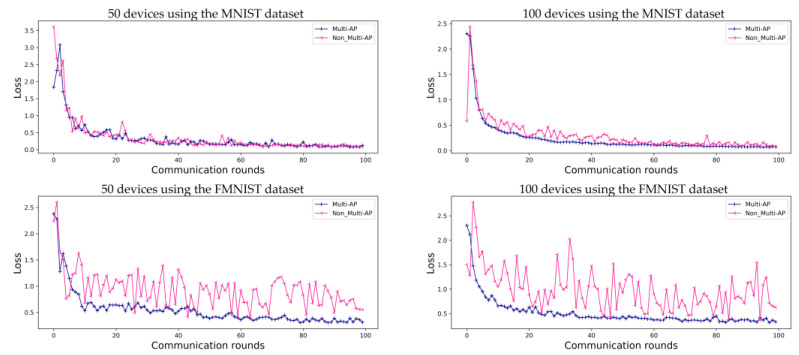
Evaluation of FL prediction accuracy loss using the MNIST and FMNIST datasets in environments with ten APs, for 100 communication rounds.

**Figure 8 sensors-22-09776-f008:**
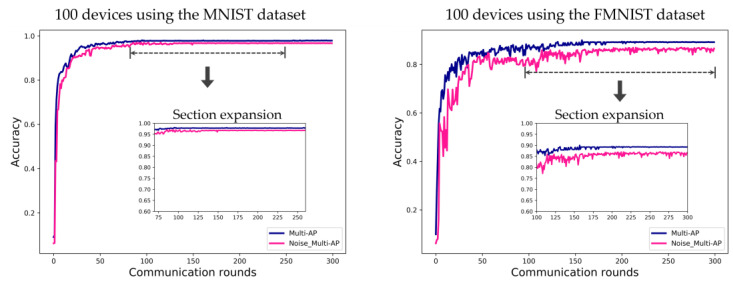
Evaluation of FL accuracy convergence using the MNIST and FMNIST datasets in environments with ten APs, for 100 communication rounds. Label ‘Multi-AP’ is the system with a good channel environment as proposed method, and label ‘Noise_Multi-AP’ means a system with a poor channel environment as the comparison.

**Figure 9 sensors-22-09776-f009:**
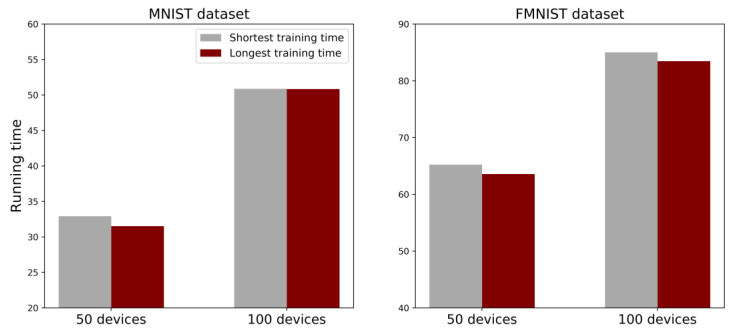
Average running time for FL using the MNIST and FMNIST datasets in environments with 50 and 100 devices in ten APs, for 100 communication rounds. The label ‘Longest training time’, as the proposed method, shows the system in which the AP with the longest training time is determined as the P-AP. The label ‘Shortest training time’, as the comparison, shows the system in which the AP with the shortest training time is determined as the P-AP.

**Table 1 sensors-22-09776-t001:** Datasets used for FL in this study.

Dataset	Training Sample	Test Sample
MNIST	60,000	10,000
FMNIST	60,000	10,000

## Data Availability

Not applicable.

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
