# Peer review of "Fairness-Based Multi-AP Coordination Using Federated Learning in Wi-Fi 7"

_sensors, 2022, doi:10.3390/s22249776_

Round 1

Reviewer 1 Report

Please refer to the peer-review results which is listed in a following file attached.

Reviewer 2 Report

This paper addresses a very interesing topic in Wi-Fi communication. However, there are minor language issues and major math issues that needs to be addressed before publication.

English

- Lots of acronyms are not defined.

- Fig and Figure are used. Please choose one format and stick to it. 

Math

- Please use a notation and description table for the symbols used in the paper.

- M is the set of AP's, with this definition d_1 to d_m in eq. 1, sounds like one device from each AP, which is not the case.

- K is actually the number of devices that participate in the FL

- N is the sum of N_k and again N is a set of {x_i, y_i}, which is two different definition,

- Energy consumption of devices can be considered to select M_AP, but the energy consumption of APs is not essential, because APs are usually connected to a power source. 

Reference

- There are few references in this work. Add recent references and some early sources in MLO

--Distributed and jamming-resistant channel assignment and routing for multi-hop wireless networks

--Beacon-based channel assignment and jammer mitigation for MANETs with multiple interfaces and multiple channels
